# Serological test performance for bovine tuberculosis in cattle from herds with evidence of on-going infection in Northern Ireland

**Lyanne McCallan**[1][☯]*, **Cathy Brooks**[1], **Claire Barry**[1], **Catherine Couzens**[1], **Fiona J. Young**[1], **Jim McNair**[1][☯][‡], **Andrew W. Byrne**[1,2][☯][‡][¤]

1 Veterinary Sciences Division, Agri-Food and Biosciences Institute, Stormont, Belfast, Northern Ireland,
2 School of Biological Sciences, Queen's University Belfast, Belfast, Northern Ireland

☯ These authors contributed equally to this work.
¤ Current address: One-Health Scientific Support Group, Department of Agriculture, Food and the Marine (DAFM), Dublin, Ireland
‡ JMN and AWB are joint senior authors on this work.
* Lyanne.McCallan@afbini.gov.uk

**Data Availability Statement:** All data was provided through the APHIS dataset, for which the data controller is DAERA. All data from which inferences were made are provided within the paper, raw test

## Abstract

The ability to accurately identify infected hosts is the cornerstone of effective disease control and eradication programs. In the case of bovine tuberculosis, accurately identifying infected individual animals has been challenging as all available tests exhibit limited discriminatory ability. Here we assess the utility of two serological tests (IDEXX *Mycobacterium bovis* Ab test and Enfer multiplex antibody assay) and assess their performance relative to skin test (Single Intradermal Comparative Cervical Tuberculin; SICCT), gamma-interferon (IFNγ) and post-mortem results in a Northern Ireland setting. Furthermore, we describe a case-study where one test was used in conjunction with statutory testing. Serological tests using samples taken prior to SICCT disclosed low proportions of animals as test positive (mean 3% positive), despite the cohort having high proportions with positive SICCT test under standard interpretation (121/921; 13%) or IFNγ (365/922; 40%) results. Furthermore, for animals with a post-mortem record (n = 286), there was a high proportion with TB visible lesions (27%) or with laboratory confirmed infection (25%). As a result, apparent sensitivities within this cohort was very low (≤15%), however the tests succeeded in achieving very high specificities (96–100%). During the case-study, 7/670 (1.04%) samples from SICCT negative animals from a large chronically infected herd were serology positive, with a further 17 animals being borderline positive (17/670; 2.54%). Nine of the borderline animals were voluntarily removed, none of which were found to be infected post-mortem (no lesions/bacteriology negative). One serology test negative animal was subsequently found to have lesions at slaughter with *M. bovis* confirmed in the laboratory.

data is available via via Figshare (https://figshare.com/s/db7f5956d1094066180a, DOI: 10.6084/m9.figshare.5705893). Additional information on these data is available from DAERA, Northern Ireland (https://www.daera-ni.gov.uk/access-information-0; daera.informationmanager@daera-ni.gov.uk) and would be subject to appropriate GDPR and Data Protection regulations (UK) in relation to individual herd keepers/herds.

**Funding:** This study was funded by the Department of Agriculture, Environment and Rural Affairs (DAERA) as part of the Evidence and Innovation Strategy under the grant "An assessment of commercially available serological tests for the detection of cattle infected with bovine tuberculosis" (grant no.: 15/3/09, Project Leaders: Dr. J. McNair and Dr. A. Byrne). The funders had no role in study design, data collection and analysis, decision to publish, or preparation of the manuscript.

**Competing interests:** The authors have declared that no competing interests exist.

## Introduction

Bovine tuberculosis is a globally distributed infectious disease. The impact of infection in cattle at the national and local level can be profound [1, 2]. For example, in Northern Ireland legislation is in place, supported by the United Kingdom and the European Union, to control this disease with the eventual aim of total eradication [3]. In practical terms, disease control across Northern Ireland is implemented through the single intradermal comparative cervical tuberculin (SICCT) test and through carcass inspection at abattoirs where cattle are slaughtered [3]. Animals identified as skin test reactors, either by standard or severe test interpretation, are removed for slaughter by compulsory order and examined post-mortem. Furthermore, all animals slaughtered at abattoirs in Northern Ireland are examined for the presence of tuberculous lesions. Clinical material collected during meat inspection is cultured for the presence of acid-fast bacteria with subsequent identification of species and strain type [4].

Despite the introduction of statutory control measures to identify and remove infected cattle, bovine TB is a persistent problem in Northern Ireland [5]. The epidemiology of disease is complicated by the presence of infection in wildlife [6, 7], and the potential confounding effects of concurrent infections on diagnostic tests [8–10]. Current diagnostic tests applied to cattle are not sufficiently sensitive to identify all infected animals and to remove them before infection is spread [11–14]. This is despite the introduction and widespread use of the interferon gamma release assay (IFNγ) [15] to augment the bovine TB testing regime and to support the front-line tests [16]. In combination, meat inspection, the skin test and IFNγ tests will identify a significant number of infected cattle, but not all [17]. It is therefore important to investigate and validate tests or improved test strategies that will broaden the capacity to identify infected animals.

The development of serology-based assays has been very useful for diagnosis where there is a Th2 type immune response. Such assays can be high throughput, relatively inexpensive and blood samples can be submitted to the laboratory a substantial time after they have been taken from the animal. However, with certain diseases a Th1 type immune responses predominates and antibody tests are largely inappropriate. This is usually the case with bovine TB when following infection, the immune response is influenced by T-cells that direct and maintain a response dominated by IFNγ release [18]. Should disease progress and the burden of infection increase then the immune response changes subtly to a Th2 type where B-cells release antibody [19]. In this situation and in the absence of cell mediated responses that can be exploited using the skin test or the IFNγ assay, an antibody assay may prove useful in the diagnosis of disease. In order to assess the role of antibody tests within a disease control programme that is already based on cell mediated responses, we instigated a study that was centred on bovine TB diseased cattle and at-risk herds. In the study reported here, we compared results from two blind tested serological tests (IDEXX *M. bovis* Ab test and Enfer multiplex serological test) with the skin test, post-mortem examination, culture confirmation and the IFNγ assay in order to define the utility of serology as a potential diagnostic test. We tested whether there was any association between test outcomes and the sex, age, and breed of animals. We also report on a case-study where one of the serological tests (IDEXX) was used in a large herd where there was a recent chronic history of bTB, and where statutory tests were failing to clear infection.

## Materials and methods

### Ethical approval

Ethical approval for withdrawal of whole blood samples was not required. Whole blood samples were drawn for bovine IFNγ testing conducted as part of the Northern Ireland TB

eradication programme (in compliance with EU Council Directive 64/432/EEC) with subsequent use in this study approved by the Department of Agriculture, Environment and Rural Affairs (DAERA) in Northern Ireland.

## Whole blood sampling

Samples intended for analysis were taken from cattle from Northern Ireland TB reactor herds eligible for inclusion in the IFNγ testing scheme operated by the Department for Agriculture, Environment and Rural Affairs (DAERA), Northern Ireland [16, 17]. Animals under six months old were excluded from IFNγ testing and therefore not included in the analysis. Individual blood samples were taken just prior to the inoculation of tuberculins on day one of the skin test and were submitted to the laboratory within 8 hours of collection. Whole blood was removed and stimulated with antigens, to be tested later for IFNγ release. Residual whole blood was centrifuged for 15 minutes to separate plasma from blood cells. Clarified plasma samples were removed individually and stored at -20°C for serological testing. Plasma samples from 407 animals positive to SICCT or IFNγ and 515 ante-mortem test negative animals (SICCT and IFNγ negative) were selected for serological testing.

## The skin test and carcass inspection at abattoir

All animals included in the study were skin tested under Annex A, Council Directive 64/432/EEC using Prionics tuberculins (PPD$_{bovis}$ and PPD$_{avium}$). Each tuberculin (0.1mL) was injected intradermally at 3000 IU (PPD$_{bovis}$) or 2500 IU (PPD$_{avium}$) on day one of the test. Skin thickness measurement, pre- and 72 hours post-injection was used to calculate increased skin thickness and to indicate the diagnostic outcome of the test. Skin test positive cattle (standard interpretation, 4mm) were submitted for slaughter at a designated abattoir in Northern Ireland (WD Meats Ltd) where carcass inspection was carried out to reveal the presence or absence of tuberculous lesions. Carcass inspection was carried out following a standardised protocol defined by DAERA with head (sub-mandibular, parotid and retro-pharyngeal), chest (bronchial and mediastinal), abdominal (mesenteric) and carcass (prescapular, popliteal, iliac and precrural) lymph nodes examined as well as the lungs, pleura and peritoneum. Tissue samples were taken from tissues with and without tuberculous-like lesions and submitted to the culture laboratory. Information pertinent to the skin test, and abattoir inspection as well as laboratory test data was recorded onto the Animal and Public Health Information System (APHIS) operated by DAERA.

## Blinded approach to laboratory tests

Sample testing was conducted using a single blind study design in which sample information, including herd number, ear tag, and statutory laboratory test results, was withheld from technical staff. This was achieved by assigning arbitrary codes to plasma samples upon collection. The arbitrary codes and corresponding sample information was stored in a database which was controlled by a senior technician. In compliance with data protection, information relating to herd keepers, herds, animals, or samples was withheld.

## The interferon gamma release assay (IFNγ test)

Whole blood samples were tested for IFNγ release using the Bovigam assay (Prionics, Switzerland) accredited by the United Kingdom Accreditation Service (UKAS). The methodology has been described previously [15]. Briefly, whole blood samples were received into the laboratory within eight hours of removal from the animal and stimulated overnight with pokeweed mitogen (2μg/ml) (internal positive control), phosphate buffered saline (nil antigen control),

$PPD_{bovis}$ (72μg/ml), $PPD_{avium}$ (36μg/ml) and ESAT-6 (0.5μg/ml). After overnight culture at 37˚C, plasma supernatant fluids were removed and stored prior to test by ELISA. The ELISA was carried out according the manufacturer's protocol with regards to reagent dilutions, incubation times and plate wash regimes. Individual sample results were accepted and recorded if reagent control and quality assurance standards were met. Those samples with Net PPDb and PPDb-PPDa optical density (OD) indexes of 0.1 or greater were positive and those less than 0.1 OD units were negative.

## Selection of serological tests

Tests to be evaluated were based on commercial availability and/or through fulfilling the requirements to test samples via a public tender established by AFBI. Two test providers were identified (see below) who satisfied the requirements.

**The IDEXX ELISA for antibodies.** IDEXX *M. bovis* ELISA kits were purchased from the manufacturer and the assay was carried out according to the manufacturer's protocol. The IDEXX ELISA is a commercially available kit. This ELISA has a 96 well microtitre plate format that detects antibodies to two *Mycobacterium tuberculosis* complex antigens (MPB70 and MPB83) known to be serological targets in *Mycobacterium bovis* infections. Briefly, plasma samples were diluted to 1 in 50 in PBS and tested in duplicate. One hundred microliters of reagents were added to wells in duplicate and incubated for 60 minutes then washed 6 times. Assay positive and negative test control reagents were used to validate each microtitre plate and provided data to calculate the test result [sample—nil / positive–nil (S/P ratio)]. Test results were interpreted as per manufacturer's instructions as follows: an S/P ratio greater or equal to 0.30 was considered positive and a ratio less than 0.3 was negative.

**The Enfer provisioned antibody assay.** An Enfer provisioned assay was carried out by Enfer staff at their Naas laboratories (Enfer ltd, Naas, Co Kildare). All tests were blinded, with no information on the epidemiological situation (e.g. within-herd prevalence) from which animals were selected provided to Enfer. It should be noted that this Enfer multiplex antibody assay is not a commercially available as a standalone kit, but testing was provided in fulfilment of commercial services as part of a commercial tender to AFBI. The basis for this assay methodology has been described previously [20]. For this study, the defined antigens used in this assay were MPB83, ESAT-6, CFP-10 and MPB70. Enfer scientific printed the bespoke multiplex according to the tender requirements, and carried out the screening, utilising bespoke software to read the multiplex plates [20]. It should also be noted that this study did not include protein fusions and cocktails, which may have been used in other studies employing the Enfer test. Plasma samples were diluted to 1 in 250 (in Enfer sample buffer A) and added to each well and incubated and agitated for 30 minutes. After washing, horseradish conjugated anti-bovine immunoglobulin was added, incubated and washed again. Substrate was added and signals were captured during a 45 second exposure stored as relative light units. The manufacturer recommends that a positive result is recorded when a minimum of any two antigens are test positive. For the purposes of this study the Enfer raw data were interpreted in two different ways. For the Enfer 2ag interpretation, a positive result was recorded if plasma samples were test positive against either MPB70 or MPB83. For the Enfer 4ag interpretation, a positive result was recorded if any two antigens, from MPB70, MPB83, ESAT-6 and CFP-10, were test positive (in line with Enfer low specificity 2ag interpretation).

## Laboratory confirmatory tests for mycobacteria

Clinical samples removed from animals at slaughter were submitted to a containment level-3 laboratory for preparation, decontamination and inoculation onto solid and liquid media.

Culture procedures at the Statutory TB Laboratories at the Agri-food and Biosciences Institute have been described extensively previously [17, 21]. Tissue structure was disrupted using either ribolysation or grinding with sterile sand in a pestle and mortar. Prior to inoculation, clinical samples were decontaminated using 5% oxalic acid for a maximum of 30 min and washed twice with sterile PBS. Samples were then inoculated onto Lowenstein-Jensen and Stonebrink slopes, as well as into Mycobacterial Growth Indicator Tubes (MGIT) containing PANTA. At 56 days post inoculation, cultures were examined for the presence of acid-fast mycobacteria and if present were further analysed using a spoligotype method [22] to identify mycobacterial species and sub-type.

## Analysis

Throughout we estimated the Area Under the receiver operator Curve (AUC) as an assessment of the ability of the serological test to discriminate between (apparent) infection states. The AUC is measured on a continuous scale from 0 to 1; an AUC of 0.5 is no better than random, with values >0.7 considered an "adequate" diagnostic [23]. Apparent sensitivity, specificity, positive predictive value and negative predictive value were calculated and reported against alternative/pseudo-gold standards of infection status. Positive Predictive Value (PPV) is the proportion of serological test positives that were positive for the comparator diagnostics (i.e. skin test, gamma interferon, visible lesion, confirmed infection, or combination thereof); Negative Predictive Value (NPV) is the proportion of serological test negatives that were negative for the comparator diagnostics. Agreement amongst tests was explored using the Kappa statistic, a kappa of 1 indicates perfect agreement, whereas a kappa of 0 indicates agreement no better than chance.

Each diagnostic was compared against the skin test (SICCT) result, IFNγ test result and post-mortem status (abattoir findings and microbiological confirmation) of the animal, giving apparent/relative performance indices. We used the definition adopted by Whelan et al. [24] to define "true" infection status. In this case, infection was defined by an animal being positive to the skin test (SICCT standard interpretation), having a visible lesion at slaughter and having a bacteriological confirmation result (positive to histology and/or microbiological culture). Being free of infection, negative animals were negative to SICCT, without lesions at slaughter and without post-mortem bacteriological confirmation. In addition, we used a combination of IFNγ, SICCT, VL and culture confirmation, to assess the relative performance of the serology tests.

The relationship between the test status and the independent variables was modelled throughout using binary logit regression models, the outcome being the binary test result ['positive' 1; 'negative' 0] for each test. A random effect for herd id (to account for potential clustering effects) was included if significant and was tested using a likelihood ratio test. We used $\chi^2$ tests and binary logit models to assess whether there was any association between animal sex, age at blood test sample, breed type (dairy production Holstein/Friesian vs. other breeds) and the probability of a positive serological test results being disclosed.

Throughout, the dataset was organised using Microsoft excel, while all statistical analysis was undertaken using Stata version 14 (Stata Corp., Texas, USA, 2015).

## Case herd study

A case study centred on a relatively large (approximately 1000 cattle over the period) dairy herd was carried out to assess the utility of antibody detection where animals were known to be infected and resolution of the problem was proving to be difficult. This particular herd had a seemingly intractable chronic bovine TB problem which originated between 2002 and 2004.

Initially, a relatively small number of bovine TB breakdowns were recorded with subsequent confirmation of infection caused by *Mycobacterium bovis*. From 2008 onward, the rate of skin test positive cattle increased significantly with a total of 148 skin test positive animals identified between December 2008 and May 2015 as well as 2 cases of lesions at routine slaughter, i.e. skin test negative cattle sent for slaughter with confirmed tuberculous lesions disclosed during carcass inspection. Ten out the 148 skin test positive animals were confirmed positive at post-mortem inspection or in the laboratory (histology or bacteriology).

Given the disease history of this herd following routine TB diagnostic investigations, and following approval from DAERA, high risk cohorts of cattle (ante-mortem SICCT negative animals housed or managed alongside TB reactors within this herd) were blood sampled and tested for the presence of antibodies to *M. bovis* using IDEXX serology (OIE approved) in 2016. The fundamental rationale was that detecting antibody in cattle that were skin test negative may indicate the presence of infection in animals that were considered to be anergic, that is, unresponsive to cell mediated tests such as the skin test and IFNγ assay. In total, 670 samples from cattle were blood sampled having been selected on the basis of being high risk cohorts of animals where the infection was most prevalent.

## Results

### Agreement and comparison

Overall, there were 922 animals with test result data; all animals had test results for IFNγ and IDEXX, 921 had SICCT, 920 had Enfer 2ag and Enfer 4ag results, while 284 animals had a post-mortem result. These animals came from 64 herds with recent bTB breakdowns, with a mean of 14.39 animals tested per herd (Median: 9.5; Std. Dev.: 13.39; Range: 1–76). The proportions of animals positive to each of the individual tests are as follows: 121/921 (13.14%) animals were SICCT positive, 365/922 (39.59%) IFNγ positive, 40/921 (4.34%) IDEXX positive, 30/921 (3.26%) Enfer 2ag positive, 13/921 (1.41%) Enfer 4ag positive, and 78/284 (27.46%) animals were found to have TB like lesions at post-mortem.

Of the animals with visible lesions found at post-mortem, the proportions deemed positive were not significantly different between the serological test types: IDEXX 10/68 (14.71%), Enfer 2ag 9/68 (13.24%), Enfer 4ag 7/68 (10.29%) (McNemar's test: Enfer 2ag vs. IDEXX: p = 0.65; Enfer 4ag vs. IDEXX: p = 0.16; Enfer 4ag vs. Enfer 2ag: p = 0.18). Similarly, there were no differences between test types, when using bacteriological confirmation as the infection status diagnostic (p>0.25).

### Serology test performance in comparison with single or combined diagnostic techniques

The relative performance of the serological tests in comparison with single ante-mortem diagnostics (Table 1), post-mortem diagnostics (Table 2) and combined tests (Tables 3 and 4) are presented below.

Relative to single ante-mortem tests (mean test prevalence 27%; Table 1), the serological tests did not disclose a high proportion of test-positive animals (mean 3% positive). This resulted in the tests exhibiting low apparent sensitivities, averaging 5.73% (range: 4.13% - 9.09%). However, the apparent specificities were always very high, with a mean of 97.82% (96.40% - 99.50%). While there was a significant positive relationship between serological test result and statutory ante-mortem outcome, the discriminatory ability of the tests were always poor (mean AUC: 0.52).

Similar results were found when post-mortem diagnostic techniques were used as the apparent infection status (Table 2). Due to the low sensitivity of the serological antibody tests,

**Table 1. The relative performance of serological tests against statutory ante-mortem tests.**

| Test type | n | Comparator | P-value | AUC | Sens | Spec | PPV | NPV | Prev. (comparator) [a] | Test prev. [b] |
|-----------|-----|-----------|---------|------|-------|--------|--------|--------|----------------------|----------------|
| ENFER 2ag | 919 | Skin test | 0.008 | 0.52 | 7.44% | 97.40% | 30.00% | 87.40% | 13% | 3.26% |
| ENFER 2ag | 920 | IFNγ | 0.009 | 0.52 | 5.22% | 98.00% | 63.30% | 61.20% | 40% | 3.26% |
| ENFER 4ag | 919 | Skin test | 0.012 | 0.52 | 4.13% | 99.00% | 38.50% | 87.20% | 13% | 1.41% |
| ENFER 4ag | 920 | IFNγ | 0.013 | 0.51 | 2.75% | 99.50% | 76.90% | 61.00% | 40% | 1.41% |
| IDEXX | 921 | Skin test | 0.008 | 0.53 | 9.09% | 96.40% | 27.50% | 87.50% | 13% | 4.34% |
| IDEXX | 922 | IFNγ | 0.091 | 0.51 | 5.75% | 96.60% | 52.50% | 61.00% | 40% | 4.34% |

Skin test: SICCT standard interpretation.

[a]Prev. (comparator): the prevalence of the respective comparator, e.g. skin test etc.

[b] Test prev.: the prevalence of the serological test being reported e.g. ENFER 2ag

the mean test prevalence was always low (mean test prevalence 4.92%) relative to the proportion of animals with lesions or post-mortem confirmed infection (mean prevalence 26%).

Using similar criteria to Whelan et al. [24] to define animals as "truly" infected and non-infected, we found that the serological tests exhibited poor sensitivity (9.09% - 13.64%; Table 3).

Utilising IFNγ test results, as an additional criterion (Table 4), suggested again that the serological tests exhibited low sensitivities, however the three serological tests achieved 100% apparent specificities.

Table 5 gives the breakdown of animal ante-mortem test results in relation to each serological test result. Overall, 8 (8/513; 1.56%), 2 (2/513; 0.39%), and 17 (17/514; 3.31%) animals were ante-mortem test negative, that were deemed serologically test positive to Enfer 2ag, Enfer 4ag and IDEXX, respectively.

Table 6 gives a breakdown of animals with post-mortem confirmed *M. bovis* infection, that were skin-test, IFNγ, or either skin-test/ IFNγ negative. Enfer 2ag and IDEXX both disclosed as positive 3/19 (15.79%) SICCT false-negative animals. The Enfer 4ag test disclosed two animals of these 19 animals as positive. However, none of the 14 post-mortem confirmed animals that were as IFNγ negative were found to be serologically positive. Overall, 6 of the animals with confirmed infection were missed by both SICCT and IFNγ tests (6/286; 2.10%), and none of these were disclosed using any of the serological antibody tests.

## Sex, age and breed associations with serological test results

There was a lack of evidence in support for an association between sex on the probability of an animal disclosing as serological positive across all tests (OR 95%CI straddled 0 for all models; p>0.05; Enfer2 ag positive: Males 3.1%; Females 3.3%; Enfer 4ag positive: Males 3.1%; Females 1.1%; IDEXX positive: Males 4.3%; Females 4.4%). Similarly, there was limited evidence of an

**Table 2. The relative performance of serological tests against statutory post-mortem diagnostic techniques.**

| Test type | n | Comparator | P-value | AUC | Sens | Spec | PPV | NPV | Prev. (comparator) | Test prev. |
|-----------|-----|---------------|---------|------|--------|--------|--------|--------|-------------------|------------|
| ENFER 2ag | 283 | Visible lesion | 0.002 | 0.55 | 12.80% | 97.60% | 66.70% | 74.60% | 28% | 5.30% |
| ENFER 2ag | 285 | Confirmed | 0.003 | 0.55 | 12.70% | 97.20% | 60.00% | 77.00% | 25% | 5.26% |
| ENFER 4ag | 283 | Visible lesion | 0.007 | 0.54 | 8.97% | 98.50% | 70.00% | 68.40% | 28% | 3.53% |
| ENFER 4ag | 285 | Confirmed | 0.004 | 0.54 | 9.86% | 98.60% | 70.00% | 76.50% | 25% | 3.51% |
| IDEXX | 284 | Visible lesion | 0.001 | 0.56 | 14.10% | 97.10% | 64.70% | 74.90% | 27% | 5.99% |
| IDEXX | 286 | Confirmed | 0.002 | 0.55 | 14.10% | 96.70% | 58.80% | 77.30% | 25% | 5.94% |

**Table 3. The relative performance of serological tests against a combination of statutory ante-mortem and post-mortem diagnostic techniques.**

| Test type | n | Comparator | P-value | AUC | Sens | Spec | PPV | NPV | Prev. (comparator) | Test prev. |
|---|---|---|---|---|---|---|---|---|---|---|
| ENFER 2ag | 187 | SICCT + VL + CONFIRM | 0.019 | 0.54 | 9.09% | 99.30% | 80.00% | 78.02% | 24% | 2.67% |
| ENFER 4ag | 187 | SICCT + VL + CONFIRM | 0.045 | 0.53 | 6.82% | 99.30% | 75.00% | 77.60% | 24% | 2.14% |
| IDEXX | 188 | SICCT + VL + CONFIRM | 0.006 | 0.56 | 13.64% | 97.92% | 66.67% | 78.77% | 31% | 6.25% |

age effect on the probability of animals disclosing with serological positive test (OR 95%CI straddled 0 for all models; p>0.08; Enfer 2ag positive vs. negative mean age (SD): 4.2 (3.1), 3.6 (2.8); Enfer 4ag positive vs. negative: 4.8 (3.7), 3.6 (2.8); IDEXX positive vs. negative mean age: 3.7 (3.1), 3.6 (2.8)). Overall, 47% of all animals were Holstein/Friesian dairy breed; 2.8% of these dairy breed animals were positive to Enfer 2ag relative to 3.7% for other breeds (Pearson $\chi^2$ (df: 1) = 0.612; P = 0.434). For IDEXX, 3.7% of dairy animals were positive, but 4.9% of other breeds were positive (Pearson $\chi^2$ (df: 1) = 0.827; P = 0.363). There was a greater difference in the proportion disclosed positive between breeds for the Enfer 4ag test, with 0.23% of dairy animals disclosing positive in comparison with 2.5% for other breed animals ((Pearson $\chi^2$ (df:1) = 8.185; P = 0.004). However, only one of the dairy animals (1/433) disclosed with a positive test.

## Case herd study

Using the manufacturer's recommended S/P ratio cut-off value of 0.3, seven samples were positive ($\geq 0.3$) and 663 samples were negative ($\leq 0.3$). Five samples were clearly positive ($> 0.3$), two samples were just above the threshold (0.340 and 0.331) and all the remaining samples were negative. However, 17 samples had S/P ratios just below the cut-off value, ranging from 0.271 to 0.113.

Following release of the serology results and discussions with the herd keeper, nine animals were voluntarily surrendered for slaughter. Seven of the nine surrendered animals were serologically positive with S/P ratios ranging from 0.331 to 1.424 and the remaining 2 animals were negative by IDEXX (S/P ratios of 0.157 and 0.223). At post-mortem examination, all cattle were designated non-visibly lesioned and clinical samples from the lung associated lymph nodes were submitted for laboratory tests. All samples were culture negative for *M. bovis*. Subsequent to this serology test-based investigation, one animal which was serology negative and submitted for voluntary slaughter, was examined and found to be visibly lesioned. Clinical samples from this animal were culture positive with *M. bovis* confirmed by spoligotype.

## Discussion

During the present study, we investigated two serological tests for their relative performance in at-risk herds in Northern Ireland. In comparison with previous work by our group [14], samples for serology testing were taken prior to the SICCT tuberculin test. This sampling approach was decided upon to allow evaluation of serology as a stand-alone test in the absence of skin testing. Whilst serology appears attractive, being relatively low cost and high

**Table 4. The relative performance of serological tests against a combination of statutory ante-mortem and post-mortem diagnostic techniques.**

| Test type | n | Comparator | P-value | AUC* | Sens | Spec | PPV | NPV | Prev. (comparator) | Test prev. |
|---|---|---|---|---|---|---|---|---|---|---|
| ENFER 2ag | 68 | SICCT + IFNγ + VL + CONFIRM | NA | 0.55 | 10.00% | 100.00% | 100.00% | 43.75% | 59% | 5.88% |
| ENFER 4ag | 68 | SICCT + IFNγ + VL + CONFIRM | NA | 0.54 | 7.50% | 100.00% | 100.00% | 43.08% | 59% | 4.41% |
| IDEXX | 68 | SICCT + IFNγ + VL + CONFIRM | NA | 0.58 | 15.00% | 100.00% | 100.00% | 45.16% | 59% | 8.82% |

**Table 5. Tabulation of the relationship between serological test results, gamma interferon (IFNγ) status by skin test status.**

|  |  | IFNγ- | IFNγ+ |  | IFNγ- | IFNγ+ |  | IFNγ- | IFNγ+ |
|---|---|---|---|---|---|---|---|---|---|
| SICCT- | Enfer 2ag- | 505 | 272 | Enfer 4ag- | 511 | 279 | IDEXX- | 497 | 274 |
|  | Enfer 2ag+ | 8* | 13 | Enfer 4ag+ | 2* | 6 | IDEXX+ | 17* | 12 |
| SICCT+ | Enfer 2ag- | 39 | 73 | Enfer 4ag- | 41 | 75 | IDEXX- | 40 | 70 |
|  | Enfer 2ag+ | 3 | 6 | Enfer 4ag+ | 1 | 4 | IDEXX+ | 2 | 9 |

* Number of ante-mortem negative animals that were serologically test positive.

throughput, this would not necessarily be true if skin testing was required prior to employing serological tests. Overall, our results suggested that the tests can achieve very high levels of apparent specificity. However, our results suggested that these tests failed to identify most animals with pathology or confirmed *M. bovis* infection post-mortem.

Research from Spain has shown when serology tests were evaluated prior to the tuberculin test, serological test performance was reduced relative to tests undertaken with samples after the tuberculin test [25, 26]. Samples taken from a cohort of animals in this Spanish study prior to skin testing suggested that the serology tests examined exhibited a sensitivity of 23.9%-32.6% (*M. bovis* Ab Test (IDEXX) & Enferplex TB assay, respectively). For animals sampled post-skin test, the beneficial anamnestic effect was most pronounced 15 days post-intradermal testing, achieving sensitivity estimates of 66.7%-85.2%. The effect was apparent by the number of animals disclosed as serology test positive when tested prior to skin testing (10.7%; 6/56), 72hrs after skin testing (7.1%; 4/56) and 15 days after testing (57.1%; 32/56).

In the current study, a small proportion of animals were disclosed as serology positive (mean 3% positive). However, during another study in Northern Ireland, we found a higher proportion of animals were disclosed as positive when prevalence was higher (86% SICCT test reactors) and testing occurred after skin testing [14]. The proportion serology positive in that cohort was 39.02–62.20% positive, with apparent sensitivities relative to post-mortem confirmed infection estimated to be 68–82%. These results suggest that maximising the beneficial effects of serology testing may occur if samples are taken after skin testing. Such boosting/priming effects have been described before in cattle in several studies [25, 27–31] and in other species also [32]. Two antigens used in the tests assessed during the present study are known to be boosted by skin testing (MPB83 and MPB70; 35). Such effects have led to some authorities to require follow-up serology testing during statutory tests, for example with camelids in Wales [32].

In the present study, a small proportion of confirmed infected (post-mortem, histology and / or bacteriology) but SICCT negative animals were identified by the serological tests (2-3/19 animals; 10.53%- 15.79%). This suggests that, in the absence of other ancillary testing,

**Table 6. Proportion of confirmed infected animals with positive serological test results, which were missed by SICCT, IFNγ, or both ante mortem bovine TB tests.**

| Confirmed infection | ENFER 2ag | ENFER 4ag | IDEXX |
|---|---|---|---|
| SICCT- (n) | 3/19 | 2/19 | 3/19 |
| (% serology positive) | 15.79% | 10.53% | 15.79% |
| IFNγ - | 0/14 | 0/14 | 0/14 |
| (% serology positive) | 0% | 0% | 0% |
| SICCT or IFNγ - | 0/6 | 0/6 | 0/6 |
| (% serology positive) | 0% | 0% | 0% |

serological tests could be useful to identify part of this subpopulation. Previous research found of 60 truly infected SICCT negative or inconclusive animals, 53 (88.3%) were disclosed as positive using a multiplex ELISA test [24]. It is hard to account for the relatively poorer detection rate in our study relative to Whelan et al. [24], but the discrepancy can partly be explained by the relatively small number of SICCT negative, *M. bovis* confirmed animals available in the present study. Employing exact binomial confidence intervals around the proportion, suggests significant uncertainty in our estimate (exact CI: 3.38% - 39.58%).

Another potential reason for the differing outcomes from this study and some other studies using the Enfer test platform, is that there was a limited set of antigens used in the current analysis, namely MPB70, MPB83, ESAT-6 and CFP10. The Enfer multiplex can detect antibody activity to 25 antigens in a single well in a 96-well plate array format [20]. However, to make cross-comparisons, only the most commonly used antigens were used during the present study. Such issues do not arise with the IDEXX *M. bovis* Ab test, as it is a standard commercial kit. Additionally, differing outcomes from this study and other studies using the IDEXX or Enfer test formats could be ascribed to the fact that this study tested plasma rather than serum, however, it should be noted both tests are marketed for use with bovine serum and plasma.

In Northern Ireland, IFNγ is routinely used in herds with problems clearing infection [16, 17]. We found in this study, that when IFNγ was used instead of, or in parallel with, SICCT, there were no additional *M. bovis* confirmed animals identified by the serological tests employed. This suggests, where both SICCT and IFNγ are used together, there may be limited opportunities to detect additional missed infected animals using serological tests. Casal et al. [26], however, suggests that in very high prevalence regions there may be value in parallel interpretation of cellular and antibody detection techniques to maximise sensitivity.

During the case study presented, few animals were disclosed as serologically positive from a large herd with a substantial chronic bTB problem. Even with liberal interpretation of the serology test (IDEXX) data, few animals were removed, and tuberculous like lesions were not observed in any of those culled nor could *M. bovis* be isolated from samples taken from these animals. One animal that was serologically tested, and found negative, was subsequently found to have visible lesions and confirmed for *M. bovis* post-mortem. This field application of the test in a particularly problematic herd appears to corroborate our other findings presented in this manuscript. However, other case-studies have highlighted benefits of serology as ancillary tests in eradicating TB. For example, a red deer herd in England with a TB outbreak was cleared of infection with the use of both tuberculin testing and serological testing over a 2-year period [33]. The authors suggest that without the additional removal of serologically test positive, the time to eradication may have been significantly increased as well as contributing to maintenance and potential transmission to local wildlife. O'Brien et al. [27] also describes a case-study in a goat herd where skin tests failed to identify all infected animals, with 6/20 slaughtered animals having visible lesions and serologically positive to six *M. bovis* antigens.

Serological tests could be strategically useful in the case of anergic animals, where advanced and generalised infection is present leading to failure to respond to SICCT due to an impaired cell mediated immunity (CMI) response [12]. However, currently there is limited data on the proportion of animals that could be deemed anergic in Northern Ireland farms. Potentially, the repeated application of SICCT testing over an animal's lifetime could lead to desensitisation [12, 34], again resulting in false negatives. When we looked at the impact of age on the probability of disclosure, we found no significant variation in our cohort. We found some weak evidence for variation in disclosure depending on breed-type, with generally Friesian/Holstein cattle exhibiting lower probability of disclosing serology positive (though this effect appeared to be only large on one of the tests, Enfer 4ag). Further research is required to ascertain whether this is a robust finding–there is significant uncertainty with the current study

given the very small numbers of animals serologically test-positive. However, previous research has suggested that there may be significant variation in *M. bovis* susceptibility and pathology across breeds [35, 36], which could be partially attributed to immunological or genetic variation [37], or other management factors.

## Conclusions

We have shown that two available serological tests, when applied to cattle populations with moderate prevalence and with samples taken prior to tuberculin testing, can exhibit limited apparent sensitivities but very high specificities. Serological tests can disclose additional test-positive animals when used in parallel with the skin tuberculin test. However, we found in this study, that when IFNγ was used instead of, or in parallel with, SICCT, there were no *M. bovis* confirmed animals identified by the serological tests employed. This suggests, where both SICCT and IFNγ are used together, there may be limited opportunities to detect additional missed infected animals via the serological tests examined when samples were taken prior to skin testing. From a perspective of a country with an ongoing extensive eradication scheme, future strategic use of serology may be limited to: 1. extreme cases of very large breakdowns within herds leading to high within-herd bTB prevalence, 2. in problem herds where IFNγ testing is unavailable, and 3. chronically infected herds where blood samples are taken after tuberculin testing to maximise sensitivity gained from any anamnestic effects.

## Supporting information

**S1 Table. Serological study raw data including IFNγ, serological, SICCT test results and abattoir findings.** 1 = positive; 0 = negative.
(XLSX)

## Acknowledgments

We thank the TB Immunology, TB Culture and TB Molecular laboratories at AFBI for their expertise in interferon gamma testing, bacteriology, and molecular confirmation techniques, respectively, as well as Jordon Graham for his assistance with capturing data from the Animal and Public Health Information System (APHIS).

## Author Contributions

**Conceptualization:** Lyanne McCallan, Fiona J. Young, Jim McNair, Andrew W. Byrne.

**Data curation:** Lyanne McCallan, Cathy Brooks, Claire Barry, Catherine Couzens, Jim McNair.

**Formal analysis:** Lyanne McCallan, Jim McNair, Andrew W. Byrne.

**Funding acquisition:** Jim McNair.

**Investigation:** Claire Barry.

**Methodology:** Lyanne McCallan, Cathy Brooks, Claire Barry, Catherine Couzens, Fiona J. Young, Jim McNair.

**Project administration:** Fiona J. Young, Jim McNair, Andrew W. Byrne.

**Supervision:** Lyanne McCallan, Fiona J. Young, Jim McNair.

**Validation:** Lyanne McCallan, Jim McNair.

**Writing – original draft:** Lyanne McCallan, Jim McNair, Andrew W. Byrne.

**Writing – review & editing:** Lyanne McCallan, Cathy Brooks, Claire Barry, Fiona J. Young, Jim McNair, Andrew W. Byrne.

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
