## [Decision Letter · Decision Letter 0]

2 Sep 2020

PONE-D-20-22608

Serological test performance for bovine tuberculosis in cattle from herds with evidence of on-going infection in Northern Ireland

PLOS ONE

Dear Dr. McCallan,

Thank you for submitting your manuscript to PLOS ONE. After careful consideration, we feel that it has merit but does not fully meet PLOS ONE’s publication criteria as it currently stands. Therefore, we invite you to submit a revised version of the manuscript that addresses the points raised during the review process.

Please submit your revised manuscript. If you will need more time than this to complete your revisions, please reply to this message or contact the journal office at plosone@plos.org. Please include the following items when submitting your revised manuscript:

We look forward to receiving your revised manuscript.

Kind regards,

Frederick Quinn

Academic Editor

PLOS ONE

Journal Requirements:

2. In your Methods section, please provide the name of the slaughterhouse where the animals were sacrificed.

Reviewers' comments:

Reviewer's Responses to Questions

**Comments to the Author**

1. Is the manuscript technically sound, and do the data support the conclusions?

Reviewer #1: Yes

Reviewer #2: Yes

2. Has the statistical analysis been performed appropriately and rigorously? 

Reviewer #1: Yes

Reviewer #2: Yes

3. Have the authors made all data underlying the findings in their manuscript fully available?

Reviewer #1: Yes

Reviewer #2: Yes

4. Is the manuscript presented in an intelligible fashion and written in standard English?

Reviewer #1: Yes

Reviewer #2: Yes

5. Review Comments to the Author

Reviewer #1: This manuscript presents results of the assessment of serological antibody tests as a potential diagnostic test of bovine tuberculosis (IDEXX M. bovis Ab test and Enfer multiplex serological test). The two tests were compared with skin test, post mortem examination, culture confirmation and the IFNγ assay. In addition, the manuscript reports the results of a case study of a large herd with a chronic history of bovine tuberculosis, where one of the serological tests was used.

the study presents interesting results that bring an added value to bovine tuberculosis diagnosis in cattle. The data were thoroughly analyzed, the results clearly presented and discussed.

Below few questions/comments:

Line 126: pokeweed mitogen is the internal positive control, right? I suggest you mention it.

Line 186-188: you mention here the collection of a selection of tissues for histological examination, can you present the results of this examination in the manuscript? If it's not possible, then there is no need to mention the tissue sample collection in formalin for histology.

Reviewer #2: To the authors.

This study (and manuscript) cover a very important topic related to the control of bovine tuberculosis in Northern Ireland. I consider the subject of testing and identification of additional infected animals (in addition to those identified by the SICCT and INF gamma) while dealing with chronically infected herds extremely relevant to the overall effort to control bovine TB in Northern Ireland.

In general, this manuscript is well written and the overall approach to analyze the collected data is appropriate. However, there are areas/sections that need clarification/correction/deletion to further enhance the presentation and interpretation of results.

Some of the material and methods must be clarified and/or expanded, and also the objective(s) of this manuscript/study needs to be revised to match and be consistent with the analyses and results presented and discussed in the manuscript.

I have inserted into the manuscript (PDF file) my specific comments, suggestions, and questions to the authors.

6. PLOS authors have the option to publish the peer review history of their article (what does this mean?). If published, this will include your full peer review and any attached files.

Reviewer #1: No

Reviewer #2: No

---

## [Author Response · Author response to Decision Letter 0]

17 Dec 2020

Response to academic editor and reviewers

Journal Requirements:

 The revised manuscript should now meet the requirements as set out in the links above.

2. In your Methods section, please provide the name of the slaughterhouse where the animals were sacrificed.

Name of slaughterhouse (WD Meats Ltd.) provided – line 108 in Manuscript file.

Title and caption added at the end of the manuscript.

Reviewer 1 comments

Line 126: pokeweed mitogen is the internal positive control, right? I suggest you mention it.

Yes, pokeweed mitogen is the positive control. Text added - line 131 in Manuscript file.

Line 186-188: you mention here the collection of a selection of tissues for histological examination, can you present the results of this examination in the manuscript? If it's not possible, then there is no need to mention the tissue sample collection in formalin for histology.

Text relating to collection of tissues for histology removed from this section of the manuscript.

Reviewer 2 comments

Abstract: purpose of the study? Case study

The case study presents information on the use of one of the serology tests (IDEXX M. bovis Ab test) in a Northern Ireland herd experiencing repeated TB breakdowns. The Department of Agriculture, Environment and Rural Affairs permitted exceptional use of this serology test in this herd in an attempt to detect additional positive animals (possibly anergic animals un responsive to SICCT or IFNƔ) previously undetected by SICCT. These details have not been included in the abstract but outlined in the materials and methods under the section titled ‘Case study herd’.

Lines 51-52: how is confounding effect used here?

Confounding effect is used here in terms of the effect of concurrent infections such as Johne’s disease and liver fluke on diagnostic tests. Text added - line 56 in Manuscript file.

Line 84: bovine TB problem – how was this defined?

“Bovine TB problem” removed. The text now reads as follows: “Samples intended for analysis were taken from cattle from Northern Ireland herds eligible for inclusion in the IFNƔ testing scheme operated by the Department for Agriculture, Environment and Rural Affairs (DAERA), Northern Ireland.” Lines 89-91 in Manuscript file.

Lines 92-93: highlighted in PDF but no comment

No changes made

Line 170: highlighted but no comment

No changes made

Lines 192-194: move to discussion

This sentence describes the AUC, and its interpretation. It should remain in the methods section. Lines 196-198 in Manuscript file.

Line 195: change was to were

Changed as requested.

Lines 197-198: post mortem results – microbiological testing / culture?

Post mortem status includes abattoir findings (lesioned / not lesioned) and microbiological confirmation. Text added – lines 207-208 in Manuscript file.

Lines 206-207: What was the outcome of these logistic regression models? How results are interpreted? Odd ratios.... for what? 

The outcome of the models is the binary test result [‘positive’ 1; ‘negative’ 0] for each test – text added in lines 218-219. 

Odds ratios removed as they are not discussed in the text.

Lines 210-211: sex, age, breed – add into study objectives

Text added to study objectives. Lines 81-82 in Manuscript file.

Line 215: consider modifying title to keep consistency with previous mentions of this part of this manuscript. "Case herd study", perhaps.

Title changed to ‘Case herd study’. Line 226 in Manuscript file.

Lines 222-223: How many of these disclosed lesions at slaughter?

Ten out of 148 positive animals were confirmed at post-mortem inspection or in the laboratory (histology or bacteriology). Lines 236-237 in Manuscript file.

Lines 226-227: This is confusing. What do you mean 'in contact', if these animals already belonged to this herd? In contact with what?

‘In contact’ means animals in contact, housed or managed alongside TB reactor cattle in the herd. Lines 239-240 in Manuscript file.

Lines 228-231: Consider moving to discussion section.

Text should stay in materials and methods under section titled ‘Case herd study’. Lines 244-246 in Manuscript file.

Lines 250-251: There was significant (p<0.001) moderate agreement between the serological tests ranging from a kappa of 0.40 (IDEXX and Enfer 4ag) to 0.55 (Enfer 2ag and Enfer 4ag). This comparison is not stated in the objective/purpose statement. Either add to it, or delete these results.

Comparison deleted from manuscript.

Line 293: Age, sex, breed associations….Were these stated in the objectives?

Text added to objectives. Lines 81-82 in Manuscript file.

Line 296: What models were used for evaluating these associations?

Odds ratios (ORs) are from logistic regression models. 

Line 303: Do all these results come from standard 2x2 tables?

Yes, Chi square tests are univariable tests (i.e. 2x2 tests).

Line 310: amend title

Amended to ‘Case herd study’ as suggested above.

Lines 311-314: move to materials and methods

Moved as requested. Lines 244-245 in Manuscript file.

Lines 324-327: Is this animal part of the 9 animals indicated above?

No this animal was not one of the 9 animals indicated above. This animal was slaughtered at a later stage, some months after the 9 animals were surrendered on the basis of the serological investigation.

Line 356: ‘see’ not needed

Removed.

Line 360: Infected..... based on what criteria? Serological test result?

Confirmed infected animals – based on post-mortem findings, histology and or bacteriology. Lines 408-409 in Manuscript file.

Line 375: M. bovis – should be in italics

M. bovis italicised. Line 423 in Manuscript file.

Line 380: ‘see’ not needed

Removed.

Line 395: from the prospective study results……. are you referring here to the other results (study) on this manuscript? I suggest to keep language consistency to ease the read

Yes this text is referring to the other results in this manuscript. Text now reads as: ‘This field application of the test in a particularly problematic herd appears to corroborate our other findings presented in this manuscript.’ Lines 440-442.

Table 1: mean not appropriate for analyses conducted. PPV /NPV need explanation / interpretation in the text

Mean removed from table 1 and other tables.

Explanation for PPV/NPV added to text. Lines 200-204.

Table 2: Odds ratio - refer to these results on the text. Prev (comparator) and test prevalence – explain, footnote.

Odds ratio removed from table 2. Prev and test prevalence explained in table 1 footnote.

Table 3: Comparator – criteria? Odds ratio - refer to these results on the text. Mean not appropriate

Odds ratio and mean removed from table 3.

Table 4: Odds ratio / P value – needed?

Removed from table 4.

Table 5: change ‘and’ to ‘by’. Numbers italicised represent ante-mortem negative animals that were serologically test positive - This explanation should not be in the title.

‘And’ changed to ‘by’.

Explanation now in table footnote.

---

## [Decision Letter · Decision Letter 1]

6 Jan 2021

Serological test performance for bovine tuberculosis in cattle from herds with evidence of on-going infection in Northern Ireland

PONE-D-20-22608R1

Dear Dr. McCallan,

We’re pleased to inform you that your manuscript has been judged scientifically suitable for publication and will be formally accepted for publication once it meets all outstanding technical requirements.

Kind regards,

Frederick Quinn

Academic Editor

PLOS ONE

Additional Editor Comments (optional):

Reviewers' comments:

Reviewer's Responses to Questions

**Comments to the Author**

1. If the authors have adequately addressed your comments raised in a previous round of review and you feel that this manuscript is now acceptable for publication, you may indicate that here to bypass the “Comments to the Author” section, enter your conflict of interest statement in the “Confidential to Editor” section, and submit your "Accept" recommendation.

Reviewer #1: All comments have been addressed

2. Is the manuscript technically sound, and do the data support the conclusions?

Reviewer #1: Yes

3. Has the statistical analysis been performed appropriately and rigorously? 

Reviewer #1: Yes

4. Have the authors made all data underlying the findings in their manuscript fully available?

Reviewer #1: Yes

5. Is the manuscript presented in an intelligible fashion and written in standard English?

Reviewer #1: Yes

6. Review Comments to the Author

Reviewer #1: (No Response)

7. PLOS authors have the option to publish the peer review history of their article (what does this mean?). If published, this will include your full peer review and any attached files.

Reviewer #1: No

---

## [Editor Report · Acceptance letter]

25 Mar 2021

PONE-D-20-22608R1 

Serological test performance for bovine tuberculosis in cattle from herds with evidence of on-going infection in Northern Ireland 

Dear Dr. McCallan:

I'm pleased to inform you that your manuscript has been deemed suitable for publication in PLOS ONE. Congratulations! Your manuscript is now with our production department. 

Kind regards, 

on behalf of

Dr. Frederick Quinn 

Academic Editor

PLOS ONE